# The Role of Physical Fitness on FRAN CrossFit® Workout Performance

**Rafaellos Polydorou [1], Andreas Kyriacou-Rossi [2], Andreas Hadjipantelis [1], Christos Ioannides [1] and Nikolaos Zaras [1,2,*]**

[1]   Human Performance Laboratory, Department of Life Sciences, School of Life and Health Sciences, University of Nicosia, Nicosia 1700, Cyprus; polydorou.r1@live.unic.ac.cy (R.P.); hadjipantelis.a@live.unic.ac.cy (A.H.); ioannides.c2@live.unic.ac.cy (C.I.)

[2]   Department of Physical Education and Sport Science, Democritus University of Thrace, 69100 Komotini, Greece; andrkyri7@phyed.duth.gr

*   Correspondence: nzaras@phyed.duth.gr; Tel.: +30-2531039718; Fax: +30-2531039623

**Abstract:** The purpose of this study was to investigate the role of physical fitness on the FRAN CrossFit® workout time-trial. Twenty male athletes were divided into a fast group (FG) and a slow group (SG) according to the median value of the FRAN time-trial. Measurements included the FRAN time-trial, body composition analysis, isometric handgrip and isometric mid-thigh pull strength, countermovement jump (CMJ), 30 s continuous jumping test ($CJ_{30}$), and one repetition maximum (1-RM) strength in the squat, thrusters, snatch, clean and jerk, and dead-lift. The FG had significantly lower body fat ($p < 0.018$), higher CMJ ($p < 0.05$), lower percentage decrement in $CJ_{30}$ height ($p = 0.023$), and higher 1-RM strength ($p < 0.05$) compared to the SG. A significant correlation was found between the FRAN time-trial with percentage body fat (r = 0.512, $p = 0.021$) and with percentage decrement in $CJ_{30}$ height (r = 0.454, $p = 0.044$). Performance in the FRAN time-trial was also correlated with CMJ variables (r ranged from 0.634 to 0.663, $p < 0.05$) and 1-RM strength (r ranged from 0.510 to 0.701, $p < 0.05$). These results suggest that the FG has a higher tolerance to fatigue and is stronger, more powerful, and has less body fat compared to the SG. Consequently, body fat, 1-RM strength, power, and anaerobic capacity may predict the FRAN time-trial in CrossFit® athletes.

**Keywords:** anaerobic test; vertical jump; power; body fat

## 1. Introduction

During the past 25 years, CrossFit® has become a well-known training method which is followed by several people who want to enhance their fitness levels. CrossFit® is a high-intensity sport which includes both aerobic and anaerobic exercises combining different athletic skills which are organized in training sessions called the workout of the day. This training scheme has several positive effects on the physiology, physical fitness, psychology, and performance of participants [1,2]. During a CrossFit® workout, the training volume and intensity can be adjusted to each individual's needs, which makes it an essential training method for participants looking for an individualized and diverse training modality [3]. CrossFit® includes several workouts with different physical fitness determinants [4–6]; however, it remains unclear what physical abilities an athlete should have developed in order to have a high performance in specific CrossFit® workouts.

One of the most well-known workouts in CrossFit® is FRAN [7,8]. FRAN consists of two multi-joint movements: front squat thrusters with a barbell of 43.2 kg and body mass pull-ups. Both exercises are performed alternately for three rounds of 21, 15, and 9 repetitions for each round. The aim is to complete the workout as fast as possible. Performance in FRAN depends largely on the tolerance of the athlete to fatigue, which may be reflected by muscle buffer capacity [9]. Muscle buffer capacity works as a system to reduce the accumulation of $H^+$ and therefore reduce the perception of fatigue during high-intensity

exercises [9]. From a practical point of view, muscle buffer capacity shows the ability of an athlete to maintain power against repeated high-intensity exercises such as the repeated sprints or the Wingate anaerobic test [10,11]. Indeed, several studies have shown that the Wingate anaerobic test may significantly predict CrossFit® performance [11,12] but not performance in FRAN [4]. More specifically, a study on 17 participants (12 males and 5 females) showed that anaerobic peak and mean power in the Wingate test were not correlated with performance in FRAN (r = −0.317 and −0.399, respectively), probably because of the level and mixed gender of the participants [4]. Although the correlation between the Wingate anaerobic test and CrossFit® performance has been previously investigated, it would be interesting to investigate the connection between continuous jumping (CJ) with FRAN. More specifically, the CJ anaerobic test refers to several maximum countermovement jumps with arms akimbo with a duration of 1 min or 30 s [13,14]. It has previously been proposed that the CJ test may be a better anaerobic test compared to Wingate for evaluating performance in trials with several stretch-shortening cycle (SSC) movements like FRAN [15]. However, whether performance in CJ may separate faster from slower athletes in FRAN may provide useful information to coaches in order to design more effective training programs and enhance performance in FRAN. In addition, a possible connection between CJ and FRAN may suggest that CJ can also be used as a laboratory tool to predict performance in FRAN. Nevertheless, such a premise needs further investigation.

Systematic CrossFit® training may lead to several significant changes in body composition, including fat mass, lean mass, and bone mineral density [16]. More specifically, 10 weeks of CrossFit®-based training may improve body composition in individuals of all fitness levels and genders [17]. Furthermore, more experienced athletes in CrossFit® possessed greater lean mass and lower body fat compared to their recreational counterparts [18], although such a comparison between faster and slower athletes in FRAN is unclear. In addition, a study on 22 (13 males and 9 females) healthy naïve CrossFit® athletes found no significant correlation between the modified FRAN and percentage body fat (r = 0.26) [6], while a study on 95 adults (including males and females) showed significant correlations between the FRAN time-trial with lean mass (ρ = −0.37 to −0.64) as well as with fat mass (ρ = 0.33 to 0.60) [19]. The scarce research data on the relationship between body composition and performance in FRAN need further investigation.

Muscle strength has a significant role in physical fitness in several sports, with stronger athletes having a greater advantage in sport-specific skills such as jumping, sprinting, and throwing [20–22]. A previous study on 15 male participants showed a significant correlation between the time-trial in FRAN and CrossFit® total (total is the sum of maximum strength in squat, strict shoulder press, and dead-lift) (r = −0.599), while 1 repetition maximum (1-RM) in back squat explained 42% of the variance in the FRAN time-trial [8]. Moreover, a study on 14 CrossFit® athletes (10 males and 4 females) found a significant correlation between CrossFit® total and FRAN time-trial (r = −0.65) [23] results, which is in line with a study on 17 CrossFit® athletes (12 males and 5 females), which also found significant correlations between the FRAN time-trial and 1-RM in squat (r = −0.644), 1-RM in dead-lift (r = −0.484), and CrossFit® total (r = −0.599) [4]. Similarly, a study on 22 novice CrossFit® athletes (13 males and 9 females) found significant correlations between performance in the modified FRAN time-trial and 1-RM in squat (r = −0.58), 1-RM in dead-lift (r = −0.57), and CrossFit® total (r = −0.61) [6]. The results from these studies suggest that 1-RM strength has a key role in the FRAN time-trial. However, Olympic lifts of the snatch and the clean and jerk [24] are regularly included in CrossFit® workouts without knowing if there is a possible connection between Olympic lifts and the FRAN time-trial. It would be interesting to investigate whether faster athletes in FRAN have greater 1-RM strength compared to slower athletes, as well as explore the relationship between the FRAN time-trial and 1-RM strength in Olympic lifts.

Therefore, the purpose of this study was to investigate the role of physical fitness on the FRAN CrossFit® workout and to examine the correlation between the FRAN time-trial with CJ, muscle strength, power, and Olympic lifts (snatch and clean and jerk) in CrossFit®

athletes. The hypothesis was that physical fitness will be significantly greater in faster athletes compared to their slower counterparts, while performance in CJ, strength, power, and Olympic lifts will be correlated with performance in the FRAN time-trial.

## 2. Material and Methods

### 2.1. Athletes

Twenty-two male CrossFit® athletes participated in this study. However, during the experimental procedure, two athletes withdrew for reasons unrelated to the experimental procedures. Thus, 20 male CrossFit® athletes (age: $30.7 \pm 4.7$ years (age ranged from 22 to 36 years), body mass: $81.2 \pm 9.3$ kg, body height: $1.76 \pm 0.07$ m) with $6.0 \pm 3.6$ years of training experience participated in this study. All athletes trained on a regular basis, with a training frequency of $4.8 \pm 0.8$ training sessions per week (maximum 7, minimum 4 training sessions per week) which classified them as recreational athletes [25]. The inclusion criteria were as follows: (a) absence of any cardiovascular problems and neuromuscular injuries, (b) systematic CrossFit® training ($\geq 4$ training sessions per week), (c) perform the FRAN CrossFit® workout at an individual pace without rest intervals, and (d) absence of any illegal drug use. Athletes were informed about the experimental procedure and signed an informed consent form. All experimental procedures were in accordance with the Declaration of Helsinki and were approved by the National Bioethics Committee of Cyprus (project number: EEBK/EΠ/2023/55).

### 2.2. Experimental Design

This was a cross-sectional study dealing with the observation of CrossFit® athletes. Measurements were performed during the preparation phase of the athletes 6–8 weeks prior to competition. Twenty CrossFit® athletes were recruited to participate in this study. Athletes followed a 5-day experimental design with 48 to 72 h of rest between testing days. During the first day, athletes visited the university laboratory where a body composition analysis and a familiarization session with the laboratory measurements were performed. On the second day, athletes visited their CrossFit® box where they performed the FRAN CrossFit® workout. Following two days of rest, athletes returned to the university laboratory for an evaluation of their maximal isometric handgrip and mid-thigh pull strength, countermovement jump (CMJ), and 30 s continuous vertical jump test ($CJ_{30}$). During the fourth and fifth days, athletes performed the 1 repetition maximum (1-RM) strength test at their CrossFit® box. Following the measurements, the athletes divided into a fast group (FG: N = 10) and a slow group (SG: N = 10) according to the median value of the time-trial in FRAN (median = 275 s) [19–22]. Those below the median were classified as "fast" and those above the median were classified as "slow". Comparisons were made between groups and correlations were made between variables.

### 2.3. Body Composition and Familiarization Session

During the first day of the study, athletes visited the university laboratory for an evaluation of their anthropometric characteristics, a body composition analysis, and a familiarization session with the laboratory performance tests. All athletes were instructed to fast for approximately 8 h and abstained from any strenuous exercise the previous day [26]. In brief, the body height of the athletes was measured on a body height measuring scale. A body composition analysis was then performed on a bioelectrical impedance scale (Tanita MC-780MA, Tokyo, Japan), which measured the fat-free mass and percentage body fat. Athletes stepped onto the bioelectrical impedance scale with light clothing and without shoes and socks and stood still for approximately 45 s. The intra-class correlation coefficient (ICC) for body mass, percentage body fat, and fat-free mass were 0.998 (95% CI: Lower = 0.998, Upper = 0.999), 0.990 (95% CI: Lower = 0.982, Upper = 0.995), and 0.965 (95% CI: Lower = 0.958, Upper = 0.987), respectively. Athletes then followed a light warm-up, consisting of a 5 min bicycle ride (at approximately 70 rpm) and static and dynamic stretching. The familiarization session included the handgrip isometric test, the isometric

mid-thigh pull dynamometer, the countermovement jump (CMJ), and the first 10 s of the $CJ_{30}$.

### 2.4. FRAN CrossFit® Workout

During the second day of the study, athletes visited their CrossFit® Box to measure the FRAN time-trial. The FRAN CrossFit® workout includes two exercises: the barbell front squat thrusters with a load of 43.2 kg and the body mass butterfly pull-ups. Both exercises were performed alternately for 3 rounds of 21, 15, and 9 repetitions for each round, starting with the barbell thrusters [8]. Athletes followed a self-selected warm-up routine, which generally included running for 5 min, stretching, and individual exercises with the barbell and on the horizontal bar. Subsequently, athletes started the FRAN time-trial and aimed to finish it as fast as possible. More specifically, the barbell and the horizontal bar were 2 m apart, while athletes had instructions to stretch their arms with the barbell above their heads during the thrusters, squat until their hips reached the same height as their knees, and pull their chin above the horizontal bar during the pull-ups. During all FRAN time-trials, one certified coach, a member of the research team, was present to evaluate the technique of the repetitions and timing of the athletes. Moreover, during the FRAN time-trial, athletes received strong encouragement from the researcher and other athletes in order to finish their workout as fast as possible. Athletes were allowed one attempt for the FRAN time-trial and their time was used in the statistical analysis.

### 2.5. Maximum Handgrip and Isometric Mid-Thigh Pull Strength

During the third day of the study, athletes visited the university laboratory for an evaluation of their maximal isometric strength, CMJ, and $CJ_{30}$. Measurements began with the isometric handgrip and mid-thigh pulls. After a warm-up on a static bicycle at 70 rpm and some dynamic stretching and body mass exercises, measurements began with the handgrip strength. Handgrip strength was evaluated using a handgrip dynamometer (Takei 5401, Digital Handgrip dynamometer, Lancashire, UK). From a standing position, with shoulders adducted and the elbow in full extension, athletes applied their maximum force and maintained it for 4 s [27]. Handgrip strength was measured in both arms, allowing three maximum efforts alternately for each hand with 30 s of rest between hands. One minute of rest was allowed between attempts. The total strength from both hands was used in the statistical analysis. The ICC for handgrip strength was 0.96 (95% confidence intervals (CI): Lower = 0.975, Upper = 0.999). Five minutes following the handgrip strength measurement, athletes stepped onto the mid-thigh pull dynamometer (Takei 5002, Analogue Dynamometer, Warwickshire, UK) for evaluation of their whole body isometric strength. Knee angle was set at approximately 130–135°, as previously described [28]. After stepping onto the dynamometer, athletes bent their knees and held the barbell with a straight back. Athletes were instructed to apply their maximum force as hard as possible and sustain it for 3–4 s. Three maximum attempts were allowed, with 2 min of rest between attempts. The highest attempt was used in the statistical analysis. The ICC for the isometric mid-thigh pull was 0.998 (95% CI: Lower = 0.998, Upper = 0.999).

### 2.6. Countermovement Jump

Five minutes after the mid-thigh pull test, athletes proceeded to the CMJ test. The CMJ was performed on the optojump (OptoJump Next, Microgate, Bolzano, Italy). The athletes performed 2 CMJs with progressively increasing intensity and, subsequently, 4 maximum CMJs with arms akimbo. During the CMJ test, athletes were instructed to jump as high as possible, while 1 min of rest was allowed between efforts. Following each CMJ, researchers informed athletes about the jump height in order to motivate them for a greater effort. From each CMJ, the jumping height, power, and power per body mass were evaluated. The CMJ with the highest height was used in the statistical analysis. The ICC for CMJ height, power, and power per body mass were 0.989 (95% CI: Lower = 0.957, Upper = 0.997), 0.980 (95% CI: Lower = 0.985, Upper = 0.990), and 0.981 (95% CI: Lower = 0.978, Upper = 0.991), respectively.

*2.7. Thirty Seconds Continuous Jump Test*

Five minutes after the CMJ, athletes performed the $CJ_{30}$ test on the optojump system (OptoJump Next, Microgate, Bolzano, Italy). Continuous jumping is an all-out anaerobic test [13] which consists of maximum continuous vertical jumps with a technique similar to CMJ. Athletes were instructed to jump as high as possible with arms akimbo, from the beginning of the test until the end of the 30 s. They also were instructed to perform a half-squat and remained at the center of the optojump photocells during the testing procedure [14,29]. During the $CJ_{30}$ test, athletes were verbally encouraged to jump as high as possible. From the $CJ_{30}$ test, the percentage decrement of jump height and power relative to body mass were evaluated. For the calculation of percentage decrement during $CJ_{30}$, the following equation was used: Percentage decrement = $-\{[1 - (\text{average of all jumps}) \times 100]/(\text{best jump} \times \text{total number of jumps})\} + 100$ [30]. The ICC for the $CJ_{30}$ test was 0.875 (95% CI: Lower = 0.750, Upper = 0.980).

*2.8. 1-RM Strength*

During the fourth and fifth days, the 1-RM strength measurement in the snatch, the clean and jerk, the thrusters, the squat, and the dead-lift were performed at the CrossFit® box. The fourth day included the snatch and the clean and jerk. All lifts were performed according to World Weightlifting Federation regulations. More specifically, athletes followed a self-selected warm-up before starting the 1-RM strength in the snatch. Athletes then performed several sets with an empty barbell (20 kg) and then 2 sets of 6 repetitions at 50% of the predicted 1-RM. They then performed 2 sets of 4 repetitions at 70% of the predicted 1-RM, 1 set of 3 repetitions at 80% of the predicted 1-RM, and 1 set of 2 repetitions at 90% of the predicted 1-RM. Subsequently, athletes had 3 maximum attempts to lift as many kilograms as possible. Fifteen minutes after the snatch, the clean and jerk 1-RM measurement followed similarly to the snatch [24]. After 72 h, athletes visited the CrossFit box® for the 1-RM strength on thrusters, squat, and dead-lift. More specifically, thrusters were performed with a deep squat and full extension of the barbell above the athlete's head, the squat was performed at the deepest point that each athlete could reach, and dead-lifts were performed with weightlifting shoes on a weightlifting platform with Olympic weights. Testing was performed similarly to the snatch, while a 15 min rest period was allowed between exercises. During all 1-RM testing, a certified weightlifting coach member of the research team was present to evaluate the validity of the lifts and encourage athletes to lift a heavier load. The ICC for 1-RM strength in the snatch, clean and jerk, thrusters, squat, and dead-lift were 0.96 (95% CI: lower = 0.81, upper = 0.99), 0.97 (95% CI: lower = 0.77, upper = 0.94), 0.93 (95% CI: lower = 0.79, upper = 0.98), 0.99 (95% CI: lower = 0.96, upper = 0.99), and 0.87 (95% CI: lower = 0.91, upper = 0.97), respectively.

*2.9. Statistical Analysis*

All data are presented as mean ± SD and were normally distributed according to the Shapiro–Wilk test. A post hoc sample power analysis showed a power of 0.750 for the 20 athletes [31]. Comparisons between the FG and SG were performed with a Student's t-test for independent samples. Hedge's *g* effect size was calculated, with the following criteria used to infer the magnitude of the difference: <0.2 (trivial), 0.2–0.5 (small), 0.5–0.8 (moderate), and >0.8 (large) [32]. For the correlation analysis, the Pearson-r coefficient was used. The reliability of all measurements was determined with a two-way random effect intra-class correlation coefficient and confidence intervals. Significance was set at $p \leq 0.05$.

**3. Results**

All athletes completed all measurements without injuries. As was expected from the experimental design, the FG was significantly faster in the FRAN time-trial compared to the SG (FG: 210.0 ± 49.9 s vs. SG: 386.1 ± 99.8 s, $p = 0.001$, $g = 2.028$). Table 1 presents the results from the anthropometric characteristics and body composition analysis of the

athletes. A significant difference was found between the FG and SG only for the percentage body fat.

**Table 1.** Comparison of the anthropometric characteristics and body composition between fast and slow groups.

|  | FG | SG | *p* | Hedge's *g* |
|---|---|---|---|---|
| Age (years) | 30.3 ±4.2 | 31.0 ± 5.4 | 0.751 | 0.131 |
| Training experience (years) | 7.0 ± 3.5 | 4.9 ± 3.6 | 0.218 | 0.518 |
| Body mass (kg) | 80.9 ± 8.3 | 81.4 ± 10.6 | 0.917 | 0.04 |
| Body height (m) | 1.77 ± 0.09 | 1.75 ± 0.06 | 0.524 | 0.264 |
| Body mass index (kg·m$^{-2}$) | 25.8 ± 1.2 | 26.5 ± 2.7 | 0.420 | 0.335 |
| Fat-free mass (kg) | 71.3 ± 8.1 | 67.9 ± 6.8 | 0.324 | 0.412 |
| Trunk muscle mass (kg) | 35.9 ± 4.0 | 34.2 ± 2.9 | 0.278 | 0.455 |
| Arms muscle mass (kg) | 8.9 ± 1.2 | 8.2 ± 1.1 | 0.244 | 0.489 |
| Legs muscle mass (kg) | 22.9 ± 2.6 | 22.1 ± 2.5 | 0.458 | 0.308 |
| Fat (%) | 11.9 ± 3.6 | 16.2 ± 3.7 | 0.018 | 1.059 |

FG = fast group, SG = slow group.

Total maximum handgrip strength was not different between groups (FG: 115.0 ± 11.9 kg vs. SG: 114.3 ± 20.2 kg, *p* = 0.929, *g* = 0.036). Although no significant difference was found for the isometric mid-thigh pulls (FG: 204.7 ± 26.4 kg vs. SG: 190.9 ± 23.4 kg, *p* = 0.232, *g* = 0.502), a moderate effect size was found between groups. In addition, results from the CMJ are presented in Table 2. The fast group had a significantly higher CMJ height and power relative to body mass compared to the slow group.

**Table 2.** Comparison of the countermovement jump and 1-RM strength variables between groups.

|  | FG | SG | *p* | Hedge's *g* |
|---|---|---|---|---|
| CMJ height (cm) | 48.5 ± 5.9 | 40.0 ± 6.0 | 0.006 | 1.252 |
| CMJ power (W) | 4553.7 ± 669.8 | 4173.1 ± 540.0 | 0.170 | 0.568 |
| CMJ power per body mass (W·kg$^{-1}$) | 56.1 ± 3.3 | 51.4 ± 3.2 | 0.005 | 1.308 |
| Snatch (kg) | 88.0 ± 11.4 | 65.2 ± 14.8 | 0.001 | 1.573 |
| Clean and jerk (kg) | 113.0 ± 17.5 | 85.7 ± 14.8 | 0.001 | 1.529 |
| Thrusters (kg) | 99.3 ± 14.1 | 74.7 ± 12.5 | 0.001 | 1.670 |
| Squat (kg) | 161.0 ± 13.9 | 131.0 ± 21.8 | 0.002 | 1.489 |
| Dead-lift (kg) | 177.5 ± 19.6 | 160.3 ± 18.1 | 0.057 | 0.827 |

FG = fast group, SG = slow group.

Moreover, the FG had a significantly lower percentage decrement in vertical jump height during the CJ$_{30}$ test (FG: −15.4 ± 3.5% vs. SG: −19.1 ± 3.1%, *p* = 0.023, *g* = 1.008) (Figure 1A). However, no significant difference was found for power relative to body mass percentage decrement between groups (FG: −15.6 ± 4.2% vs. SG: −18.6 ± 4.1%, *p* = 0.120, *g* = 0.662), though a moderate effect size was observed (Figure 1B). Results from the 1-RM strength measurements are presented in Table 2. Significant differences were found for almost all strength exercises between the FG and SG, while a large effect size was observed for the 1-RM strength in dead-lift.

A correlation analysis including all athletes (N = 20) revealed significant relationships between the FRAN time-trial with percentage body fat (r = 0.512, *p* = 0.021) (Figure 2A), as well as with the percentage decrement of height during the CJ$_{30}$ test (r = −0.454, *p* = 0.044) (Figure 2B). Furthermore, the time-trial in FRAN was significantly correlated with 1-RM strength in snatch (r = −0.583, *p* = 0.007), clean and jerk (r = −0.561, *p* = 0.011), thrusters (r = −0.510, *p* = 0.022), and squat (r = −0.701, *p* = 0.001). In addition, the FRAN time-trial was also correlated with CMJ height (r = −0.634, *p* = 0.003) and CMJ power relative to body mass (r = −0.655, *p* = 0.002).

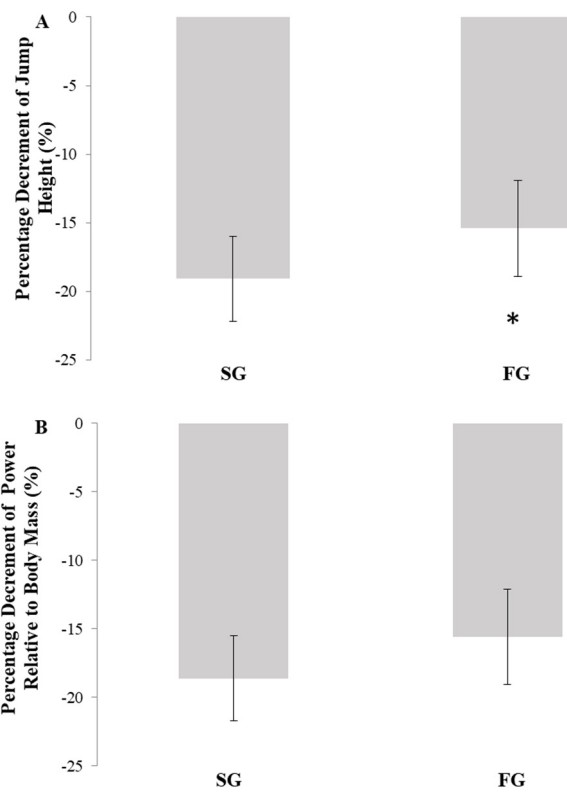

**Figure 1.** Results from the CJ$_{30}$ test. (**A**) Percentage loss of vertical jump height, (**B**) percentage loss of power relative to body mass. * *p* = significant difference between fast group (FG) and slow group (SG).

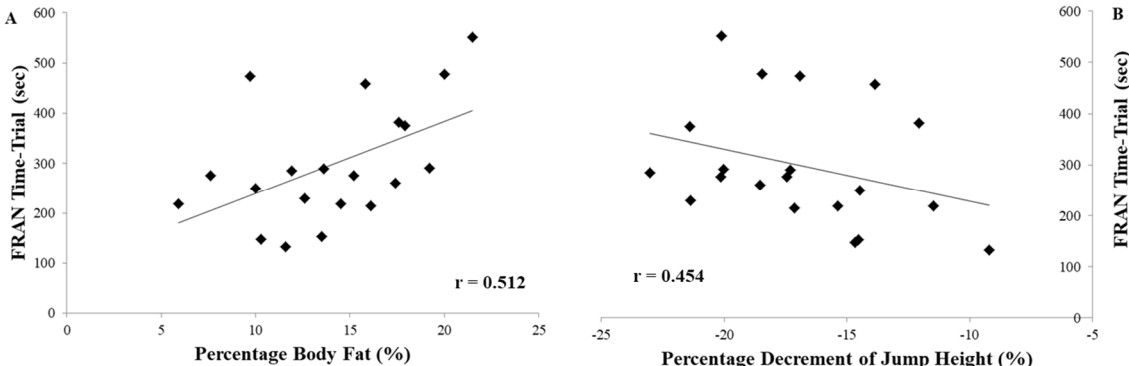

**Figure 2.** Correlation scatterplots between (**A**) performance in the FRAN time-trial and percentage body fat and (**B**) performance in the FRAN time-trial and percentage loss of jump height during the CJ$_{30}$.

## 4. Discussion

The purpose of this study was to investigate whether faster and slower athletes in the FRAN time-trial have significant differences in physical fitness as well as examine the correlations between physical fitness and the FRAN time-trial. The main finding of this study was that the FG had lower percentage body fat, lower percentage decrement in CJ$_{30}$ height, greater power per body mass in CMJ, and higher 1-RM strength compared to the SG. Faster athletes were able to complete the FRAN time-trial with higher physical fitness and lower body fat, while they had a greater tolerance to fatigue, as found from the CJ$_{30}$ test. In addition, percentage height decrement during CJ$_{30}$ was significantly correlated with the FRAN time-trial, which shows the link between resistance to fatigue and the FRAN time-trial. Similarly, percentage body fat was positively correlated with the FRAN

time-trial, which shows that body fat may be a restrictive factor for high performance in FRAN. In addition, variables from CMJ and 1-RM strength were significantly correlated with the FRAN time-trial.

One main finding of the present study was that the FG had a significantly lower percentage decrement in jump height during the $CJ_{30}$ compared to the SG, while a moderate effect size ($g = 0.662$) was found for power decrement. The $CJ_{30}$ is a reliable test and may provide useful information regarding the fatigue tolerance as well as the anaerobic profile of the athletes [14,15]. However, according to the author's knowledge, this is the first study to evaluate $CJ_{30}$ in CrossFit® athletes; thus, the interpretation of the results should be performed with caution. CrossFit® training is designed with high-intensity workouts which may result in a significant increase in muscle buffer capacity [9]. Indeed, acute and long-term training studies have shown that systematic high-intensity exercise may lead to a significant increase in maximum oxygen consumption and lactate threshold, similar to continuous exercise, but may result in greater muscle buffer improvements [9,10]. Although no significant difference was observed for the training experience between groups, it seems that the FG had a better buffering capacity [33], which may lead to a higher performance in FRAN. These results are further reinforced by the significant although moderate correlation between the FRAN time-trial and percentage decrement in $CJ_{30}$ height. Previous studies have shown significant correlations between muscle buffer capacity and repeated sprint ability [10]. It can be hypothesized that one possible mechanism that leads to a faster performance in FRAN is the higher muscle buffer capacity of the FG, which may result from the systematic higher intensity exercise that these athletes are following during training. A recent study confirmed that FRAN is a physically demanding workout which recruits energy from both aerobic and anaerobic mechanisms, leading to a significant reduction in muscle function [34]. However, no lactate samples or muscle biopsies were obtained in the current study. In addition, several studies have shown that the Wingate anaerobic test may significantly predict CrossFit® performance [11,12] but not the time-trial in FRAN [4]. However, from a biomechanical perspective, it seems that $CJ_{30}$ might be closer to front squat thrusters and pull-ups compared to Wingate, mainly because CrossFit includes SSC movements [15,29]; therefore, the assessment of anaerobic capacity in CrossFit® is suggested to be performed with CJ. Although this explanation seems rational, more studies are needed to investigate the possible link between $CJ_{30}$ and performance in CrossFit®.

An important finding of this study was that athletes in the FG had significantly lower percentage body fat compared to athletes in the SG. Studies support that systematic CrossFit® training leads to a significant reduction in percentage body fat, while it can significantly improve health-related fitness factors [1,18,35]. Indeed, a recent study investigated the body fat differences between advanced and recreational CrossFit® athletes and found that advanced athletes had lower percentage body fat compared to their recreational counterparts [18]. These results are strengthened by the large positive correlation between the FRAN time-trial and percentage body fat found in the current study. Although body fat may be a limiting factor of performance in the FRAN time-trial, other factors, like the technique of the exercises, the neuromuscular condition of the athlete during the test, or the training experience of the athlete, may influence time-trial performance in FRAN. Nevertheless, the results of the current study may suggest that athletes should focus on reducing percentage body fat to enhance their performance in the FRAN time-trial.

Significant differences were observed for CMJ between the FG and SG. More specifically, the FG had greater CMJ height and power relative to body mass compared to the SG. A study on wrestlers showed that systematic training with the CINDY workout may lead to a significant increase in CMJ height compared to wresting practice alone [36]. In addition, a study which compared high and low performers in CrossFit® found significant differences in squat jump but not in CMJ in contrast to the findings of the present study, probably due to the different level of the participants [5]. However, CMJ height was significantly correlated with overall performance in CrossFit® with workouts which included exercises like thrusters, squats, or rope double-unders [5]. Similarly, the results from the

current study showed significant correlations between all CMJ variables and the FRAN time-trial. Consequently, performance in CMJ combined with $CJ_{30}$ may provide useful information to coaches regarding the lower body power condition, as well as the athletes' anaerobic capacity.

Previous studies have shown that stronger athletes may have a greater benefit with sport-specific skills compared to their weaker counterparts [20–22]. Likewise, in the present study, the FG had significantly greater 1-RM strength compared to the SG, although no differences were observed for handgrip and isometric mid-thigh pull tests. A previous study showed that more experienced CrossFit[®] athletes had greater 1-RM strength in the bench press; however, no difference was observed for deep full squat [5], probably because of the different level of these athletes. We can hypothesize that athletes in the FG had a greater benefit from their increased 1-RM strength, which may partially lead to a higher time-trial performance in FRAN. Several studies on CrossFit[®] athletes have found significant correlations between the FRAN time-trial with 1-RM strength and CrossFit[®] total [4,6,23]. However, these studies included both male and female athletes, which might influence the correlation results. A recent systematic review revealed moderate-to-high correlations between 1-RM strength in the back squat and CrossFit[®] performance [37], results that are in line with the findings of the present study, since the correlation between the FRAN time-trial and 1-RM squat (r = −0.701) was the highest correlation found, compared to other 1-RM strength exercises. Consequently, 1-RM strength, especially in squat, seems to be a good predictor of the FRAN time-trial in CrossFit[®] athletes. However, coaches should focus not only on increasing strength in squat but in thrusters [8] and Olympic lifts as well. The findings of the current study support that 1-RM strength in the snatch and clean and jerk was correlated with the FRAN time-trial. Both Olympic lifts are considered to be highly technical multi-joint exercises which activate the entire neuromuscular system [24]. Therefore, practicing Olympic lifts may induce higher performance enhancement in CrossFit[®] workouts.

This study has some limitations. The small sample size in each group might prevent generalization of the results. Male athletes participated in the study; thus, more studies are needed to investigate differences in the physical fitness of female athletes. There were no lactate samples, muscle biopsies, or electromyography signals recorder, which might have provided a better explanation into the difference between the FG and SG. Although body fat was significantly different between the FG and SG, more studies are required to better understand the physiological differences between faster and slower athletes, not only in FRAN but in other workouts as well. Moreover, several weaknesses may arise from the current cross-sectional study; thus, more research is required with cross-over or interventional studies.

## 5. Conclusions

Faster CrossFit[®] athletes in FRAN may benefit from their lower body fat and percentage decrement in $CJ_{30}$ height, as well as from their higher power output and 1-RM strength, compared to slower athletes. Percentage body fat, percentage decrement in $CJ_{30}$ height, power, and 1-RM strength may predict faster times in FRAN. Consequently, CrossFit[®] programs should focus not only on the development of endurance and stamina but in power and maximum strength as well. In addition, CrossFit[®] athletes should regularly monitor their body composition and aim to reduce their percentage body fat, which seems to be a predictive factor of the FRAN workout.

**Author Contributions:** Conceptualization, R.P. and N.Z.; Methodology, R.P., A.K.-R., A.H., C.I. and N.Z.; Investigation, R.P., A.K.-R. and A.H.; Validation, R.P., A.H. and C.I.; Resources, R.P. and N.Z.; Formal analysis, R.P. and N.Z.; Writing—Original Draft Preparation, R.P., A.K.-R., A.H. and C.I.; Writing—Review and Editing, R.P., A.K.-R., A.H., C.I. and N.Z.; Supervision, N.Z. All authors have read and agreed to the published version of the manuscript.

**Funding:** This research received no external funding.

**Institutional Review Board Statement:** This study was conducted according to the guidelines of the Declaration of Helsinki and was approved by the National Ethics Committee of Cyprus (protocol code EEBK/EΠ/2023/55, 8 December 2023).

**Informed Consent Statement:** Informed consent was obtained from all subjects involved in this study.

**Data Availability Statement:** The raw data supporting the conclusions of this article will be made available by the authors on request.

**Acknowledgments:** The authors express their gratitude to those athletes who participated in this study.

**Conflicts of Interest:** The authors declare no conflicts of interest.

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
