# Peer review of "The Role of Physical Fitness on FRAN CrossFit® Workout Performance"

_applsci, doi:10.3390/app14083317_

Round 1
Reviewer 1 Report
Comments and Suggestions for Authors
I don't like the term WOD to indicate a part of the training session. I think the more appropriate term would be "workout". The WOD should be used for a warm-up, skill and MetCon session.
Another issue, which could be reported in the limitations, was the design of the study with 5 consecutive days of non-randomized testing. This could generate some discussion about the order used.
New studies regarding the Fran workout should be cited: Rios et al., (2024; PMID: 38194958), Silva de Souza et al. (2024; PMID: 38194958), Rios et al., (2023; PMID: 37225165), and Mangine et al. (2022; PMID: 35721570).
Author Response
I don't like the term WOD to indicate a part of the training session. I think the more appropriate term would be "workout". The WOD should be used for a warm-up, skill and MetCon session.
Response
We thank the reviewer for the comment. We changed WOD to workout throughout the manuscript.
Another issue, which could be reported in the limitations, was the design of the study with 5 consecutive days of non-randomized testing. This could generate some discussion about the order used.
Response
We thank the reviewer for the intriguing comment. There was a 48-72h of rest between measurements. In addition, the measurements were designed in this specific order in an attempt to produce the lower possible fatigue to athletes. We clarify this in the methods section.
New studies regarding the Fran workout should be cited: Rios et al., (2024; PMID: 38194958), Silva de Souza et al. (2024; PMID: 38194958), Rios et al., (2023; PMID: 37225165), and Mangine et al. (2022; PMID: 35721570).
Response
We thank the reviewer for pointing these relative references to our attention.
Reviewer 2 Report
Comments and Suggestions for Authors
1. National bioethics committee of Cyprus 106 (project number: ΕΕΒΚ/ΕΠ/2023/55). Is it IRB(Institutional Reviwer board)?
2. Giving the full name for CrossFit® is FRAN.
3. What is a FRAN? Is it kind of crossFit or another methods?
It is not clear.
4. The most problem with this study is the study design. Cross-sectional study designs don't work. It should be validated with a cross-over study or intervention study.
Author Response
Reviewer 2
- National bioethics committee of Cyprus 106 (project number: ΕΕΒΚ/ΕΠ/2023/55). Is it IRB(Institutional Reviwer board)?
Response:
We thank the reviewer for the question. This is the National Bioethics Committee where our study was approved; it is not from our University. In Cyprus we are taking bioethical approvals only from this committee.
- Giving the full name for CrossFit® is FRAN.
Response:
We thank the reviewer for the question. FRAN is the name of the workout we investigated which is included in CrossFit.
- What is a FRAN? Is it kind of crossFit or another methods?
It is not clear.
Response:
We thank the reviewer for the question. As we mentioned inside the manuscript, FRAN is a CrossFit workout which includes two exercises: thrusters and pull-ups. Athletes perform three sets accounting for 21, 15 and 9 repetitions each, and are trying to complete the workout as fast as possible. FRAN is also included inside the CrossFit method.
- The most problem with this study is the study design. Cross-sectional study designs don't work. It should be validated with a cross-over study or intervention study.
Response:
We thank the reviewer for the comment. We agree with the reviewer, however, in order to reach the cross-over and intervention study designs we must start with the cross-sectional despite the weaknesses that arising from these kinds of designs. In addition, we added the reviewer comment into the limitations paragraph.
Reviewer 3 Report
Comments and Suggestions for Authors
Thank you for the opportunity to review your study. While I think the findings could be interesting, I have some serious concerns regarding the derivation of the study rationale in the introduction, parts of the study methods and especially regarding the interpretation of the findings. My concerns are provided in the following:
The start of the introduction reads fluently. One aspect which could be addressed: in line36/37 you refer to increased injury risk, if the WOD is not appropriately designed. What is with injury risk of high complex movements such as the snatch, clean or deep squat in untrained participants? Are there references that could be included?
Lines 53 – 63: The transition from the wingate test to the jump test is quite confusing. Furthermore, if you want to make a statement about the buffer capacity, you should test buffer capacity. Otherwise references must be included stating this capacity would limit wingate as well as CJ test to the same extent.
Line 61: I do not see the SSC in the fran? Neither the front squat nor the pull up should involve SSCs. I know the horrible cross fit pull up execution. However, there you get a standardization limitation; I will refer to that in later comments
Lines 61 – 63: Which training recommendations could be improved in case you would know that the CJ test would be correlated with the time necessary to accomplish FRAN?
Lines 67/68: 15males + 5 females = 17 participants? Even though my maths skills might be outdated, a clarification on this would be great.
Lines 71 – 73: It seems that this reference in the introduction is poorly connected to the study?
Lines 65 – 85: This section needs a clearer red line to derive the research question. It seems you state random cross fit studies you found in literature.
From the current version of the Introduction I am not clear why the Methods you describe in the following were used. I only know that there might be a correlation between wingate and cross fit performance and there might be a correlation between the 1RM squat and cross fit overall performance. The study rationale is pretty unclear to me. Stating that it might be interesting to investigate the influence of the CJ, because there is a correlation between wingate and FRAN is a weak rationale for a study.
Methods
Line 112 – 113: you did not derive body composition measurements. In accordance with my previous comments outlining an unclear methods derivation from the introduction I would recommend to revise the first paper part so that it is clear after reading the Introduction to every reader it is clear: Now we will find the description of e.g. the CMJ, 1RM Squat test and CJ test.
When reading the methods section it is not clear to me why you measure body compostion? This was not mentioned before. The same is true for isometric handgrip strength and the isometric mid thigh pull. It seems you are fishing for significant correlations
Lines 140 – 153: To get you right: the athletes went to the cross fit box and performed the FRAN? Any standardization? Any supervision? Any requirements that must be fulfilled in the squat? I know the poor movement exection in many cross fit boxes very well. Most of these would not be adequate to take part of a study. Studies should use highly standardized and objective methods instead of doing anything and performing random testing. I do not want to say that you did this, it is just in general a problem in current studies I found in literature and want to make this point so that this impression is counteracted by a clear description of the quality standard in data collection you adhered to. This point is not clear from the description and a revision is necessary.
Lines 154 – 174: I did this point before, it is not clear why you tested handgrip and mid tigh pull isometric strength?
Lines 175 – 186: Same here: The CMJ is no part of the FRAN. Why did you test it?
Lines 200 – 218: The description should be improved. What was the squatting depth? How was the trial rated success? Femur position, ankle position, back position? Hip position? Were there two independent raters that classified the trial? Which requirements must be fulfilled to rate the trial successful? The weight over the head (Snatch/Jerk?) or were there further requirements such as no knee collapse, no ankle collapse, no buttwink?
Results and Discussion
You just correlated FRAN with the remaining test items. It figured out in this section that there would be interpretation limitations in the discussion. Even though there were or not were correlations, you did not check the influence of 1RM on jumping performance ? I have serious concerns regarding the physiological explanation regarding the increased jumping height and buffer capacity, you hypothesized in the introduction to explain success in the FRAN.
No Discussion was performed on the influence of maximal strength on strength-endurance capcity. If my maximal strength is very high, the used percentage strength value (e.g. 160kg you mentioned in the faster group) is smaller compared to the percentage of weaker athletes. Thus, it can also be hypothesized that they will show reduced fatigue and reduced accumulated lactate. Strength endurance performance is not only determined by the capacity to eliminate lactate, or resistvagainst high acidosis. If the needed percentage of the muscle does not trigger the anaerobic system, as the performance can be covered by the aerobic system no or lower lactate accumulation will arise. This can be explained by higher maximal strength. The Discussion (similar to the introduction) seems not physiologically guided, but accumulates several studies from the literature with questionable relevance for the study and needs a revision.
Additionally:
Lines 313 – 323: A descriptive approach and correlation between body fat and performance is no causal relationship. Just because athletes with lower body fat showed better performance, it is unvalid to state that the body fat will be the factor that caused this higher performance. Otherwise, every marathon running athlete with 2% body fat would outperform strength athletes.
Lines 324 – 336: Same is true for CMJ height. Ignoring that CMJ and 1RM max strength might be highly correlated as well, as maximum strength commonly explains a large percentage of variance in the CMJ jump, using correlation coefficients for such statements in this section are very questionable.
Lines 344 – 349: you state mixed sex distribution in previous studies as a limiting factor. Why? Does sex make the difference, if any exists? Hormonal status (there are women with high testosterone levels and men with low levels, other hormones (HGH; Insulin etc. with anabolic effects are present in both sexes). Did any study that performed sex difference analyses controlled hormonal differences? Just sex seems a week explanation, especially since there are numerous articles that show no significant difference in strength training adaptations between males and females.
Lines 351 – 356: The practical recommendations you state below might be correct (personally I would always agree with increasing 1Rm in the squat and focusing on Olympic lifts would be an essential part of every performance orientated or health related training routine), however, these recommendations might not be very well related to you study and it seems unvalid to state these based on your results. Please revise.
I think there is much more potential in your data that should be outlines. Please increase the red line in the Introduction and Discussion by avoiding stating random crossfit studies, but focus the Introduction on the studies objective you investigated in the following. I would recommend to further include correlations between maximal strength and the other parameters (each parameter with each other) to get results for the Discussion that provide additional value, as you might not have to focus on correlations between CMJ and FRAN; but could also discuss the relationship of maximal strength in CMJ and FRAN together to formulate the outlook for further studies that should investigate the cross fit profile and relationships between different motoric functions, as well as checking the relevance of e.g. a deep squat training (lower extremity maximal strength), anaerobic training to resist against high lactate acids or buffer capacity, or isometric strength (IMTP) on specific cross fit performance.
While results might be interesting in general, some more work has to be performed before I can recommend the publication of this manuscript. I hope some comments are helpful for your revisions. Looking forward to the revised version. Thank you very much for your work.
Comments on the Quality of English Languagecan be improved, some words should be revised. Overall understandable.
Author Response
Reviewer 3
Thank you for the opportunity to review your study. While I think the findings could be interesting, I have some serious concerns regarding the derivation of the study rationale in the introduction, parts of the study methods and especially regarding the interpretation of the findings. My concerns are provided in the following:
Response:
We would like to thank the reviewer for taking the time to review our manuscript. We appreciate the efforts in providing us with comments that will enhance the quality of the paper and the message we are trying to deliver to the readership. We hope our replies and amendments are of satisfaction.
The start of the introduction reads fluently. One aspect which could be addressed: in line36/37 you refer to increased injury risk, if the WOD is not appropriately designed. What is with injury risk of high complex movements such as the snatch, clean or deep squat in untrained participants? Are there references that could be included?
Response:
We thank the reviewer for the questions. Indeed, snatch, clean & jerks as well as deep squats are increasing the injury risk not only in untrained participants but in trained athletes as well. And it is not only the weightlifting derivatives that increase the injury risk, there also the gymnastics and callisthenic exercises that may contribute to this factor. However, the purpose of the study is not to focus on the injury risk that arises from crossfit but on performance. In addition, the specific sentence was deleted according to the comments of the 5th reviewer.
Lines 53 – 63: The transition from the wingate test to the jump test is quite confusing. Furthermore, if you want to make a statement about the buffer capacity, you should test buffer capacity. Otherwise references must be included stating this capacity would limit wingate as well as CJ test to the same extent.
Response:
We thank the reviewer for the comment. We can see the problem in the transition from the Wingate to CJ. We tried to balance the difference between the two. Although we have not measured muscle buffer, a recent study showed that the FRAN workout is a physically demanding activity that recruits energy from both aerobic and anaerobic systems (doi: 10.1123/ijspp.2022-0411). Moreover, a relative study which compared the Wingate and the 30sec CJ showed that CJ had greater mechanical efficiency and induced lower fatigue index compared to Wingate (https://doi.org/10.1123/ijspp.2020-0669). Consequently, we cannot state that CJ may induce to the same extend performance since the different locomotion between Wingate and CJ.
Line 61: I do not see the SSC in the fran? Neither the front squat nor the pull up should involve SSCs. I know the horrible cross fit pull up execution. However, there you get a standardization limitation; I will refer to that in later comments
Response:
We thank the reviewer for the comment. We agree with the reviewer regarding the presence of SSC during FRAN. However, we refer to the SSC as a slow continuous alternation of muscle contractions from concentric to eccentric and not as a fast SSC frequently used (https://doi.org/10.26773/mjssm.240308).
Lines 61 – 63: Which training recommendations could be improved in case you would know that the CJ test would be correlated with the time necessary to accomplish FRAN?
Response:
We thank the reviewer for the question. This is the hypothesis for the second research question. We can only recommend to coaches and athletes that training should focus on improving anaerobic capacity and perhaps the tolerance of athletes in fatigue in order to enhance performance in FRAN. In addition, CJ may also been used as a tool to predict performance to athletes with different levels of technical skills in FRAN.
Lines 67/68: 15males + 5 females = 17 participants? Even though my maths skills might be outdated, a clarification on this would be great.
Response:
We thank the reviewer for pointing this to our attention. We believe that our response would be much worse than the reviewer’s outdated maths since the study of Leitao et al. included only male athletes. We apologise for the misprint.
Lines 71 – 73: It seems that this reference in the introduction is poorly connected to the study?
Response:
We thank the reviewer for the comment. The reference was deleted.
Lines 65 – 85: This section needs a clearer red line to derive the research question. It seems you state random cross fit studies you found in literature.
Response:
We thank the reviewer for the suggestion. The selection of reference is not random. We selected each reference because of the correlations between FRAN and 1-RM strength. However, we made some changes in order to better present the research question of this particular paragraph.
From the current version of the Introduction I am not clear why the Methods you describe in the following were used. I only know that there might be a correlation between wingate and cross fit performance and there might be a correlation between the 1RM squat and cross fit overall performance. The study rationale is pretty unclear to me. Stating that it might be interesting to investigate the influence of the CJ, because there is a correlation between wingate and FRAN is a weak rationale for a study.
Response:
We thank the reviewer for the efforts to improve the clarity of our introduction. We are not focusing only on the correlational results. We are also making a comparison between athletes and this provides useful information to the readers. However, we hope that the changes we have made in the text combined with the positive comments of the reviewer have improved the quality and clarity of the intro.
Methods
Line 112 – 113: you did not derive body composition measurements. In accordance with my previous comments outlining an unclear methods derivation from the introduction I would recommend to revise the first paper part so that it is clear after reading the Introduction to every reader it is clear: Now we will find the description of e.g. the CMJ, 1RM Squat test and CJ test.
Response:
We thank the reviewer for pointing this to us. We agree with the reviewer. We have now revised the intro and discussed the role of body composition on FRAN time-trial.
When reading the methods section it is not clear to me why you measure body compostion? This was not mentioned before. The same is true for isometric handgrip strength and the isometric mid thigh pull. It seems you are fishing for significant correlations
Response:
We thank the reviewer for the question. We have now added a paragraph in the intro trying to present the necessity to investigate the role of body composition in FRAN and generally in CrossFit. In addition, we hypothesized that there might be differences between faster and slower athletes. In line with the first hypothesis of the study, we added two laboratory strength tests in order to measure with greater accuracy muscle strength, although the results showed poor outcomes. However, these measurements where part of the laboratory protocol, while readers may find it interesting for future studies.
Lines 140 – 153: To get you right: the athletes went to the cross fit box and performed the FRAN? Any standardization? Any supervision? Any requirements that must be fulfilled in the squat? I know the poor movement exection in many cross fit boxes very well. Most of these would not be adequate to take part of a study. Studies should use highly standardized and objective methods instead of doing anything and performing random testing. I do not want to say that you did this, it is just in general a problem in current studies I found in literature and want to make this point so that this impression is counteracted by a clear description of the quality standard in data collection you adhered to. This point is not clear from the description and a revision is necessary.
Response:
We thank the reviewer for the intriguing comment. As we are involved in CrossFit, we are aware of the poor technique and cheating methods some athletes use to gain an “advantage” during training. However, in our study all of the athletes had previously participated in national and international competitions. So, they had a decent competition experience. Also, performance in FRAN was supervised by one certified coach who was member of the research team. During the same time the crossfit coach was present to assist with the rep counting and technique execution. Unfortunate, it was difficult to standardise the movements by using tools to monitor the depth of the squat. It would also be dangerous for the athletes since they were not familiar with this kind of monitoring. We only gave strict instructions regarding the execution of the exercises as to stretch their arms during the thrusters, squat until their hips reach the same height of their kness, and pull themselves above the horizontal bar during the pull ups. We also standardised the distance between the barbell and the bar which were two meters apart. We added some more details in the paragraph.
Lines 154 – 174: I did this point before, it is not clear why you tested handgrip and mid tigh pull isometric strength?
Response:
We thank the reviewer for the question. Isometric handgrip and mid-thigh pulls strength were part of the laboratory protocol and measured as an accurate index of maximum strength under laboratory conditions. These findings can also be used as reference values for these athletes. Unfortunately, our study did not found any difference between fast and slow athletes as well as any significant correlation with performance. However, readers may use this information for future research.
Lines 175 – 186: Same here: The CMJ is no part of the FRAN. Why did you test it?
Response:
We thank the reviewer for the question. There are strong evidences that there is a significant relationship between CMJ and performance in weightlifting derivatives like the thrusters. In our study this correlation was r=0.645 (p=0.002). In addition, CMJ as an index of lower body power may be a useful tool for coaches. Finally, a previous study compared the CMJ and SJ between high and low performers and made correlations with 5 different workouts (doi:10.3390/ijerph17103699). It would be interesting to know that an easy to use power test may predict performance in FRAN.
Lines 200 – 218: The description should be improved. What was the squatting depth? How was the trial rated success? Femur position, ankle position, back position? Hip position? Were there two independent raters that classified the trial? Which requirements must be fulfilled to rate the trial successful? The weight over the head (Snatch/Jerk?) or were there further requirements such as no knee collapse, no ankle collapse, no buttwink?
Response:
We thank the reviewer for the intriguing comments and questions regarding the evaluation of performance in weightlifting. During all weightlifting strength measurements one certified weightlifting coach (member of the research team) was presented together with the certified coaches of each crossfit box. We followed the international rules of the WWF for the Snatch and the C&J, while for squat, athletes were tested with deep squat technique. In addition, the thruster was performed with deep squat and the dead-lift with weightlifting shoes and Olympic weights. We added some more details regarding the execution of the exercises. Unfortunately, we were not able to measure all these positions that the reviewer reasonably noted in the comment. We are a very experienced group in weightlifting, thus we helped all athletes to finish the 1-RM strength measurements with no injuries and with the highest technical execution during all exercises. We hope that our revisions inside the text will please the reviewer.
Results and Discussion
You just correlated FRAN with the remaining test items. It figured out in this section that there would be interpretation limitations in the discussion. Even though there were or not were correlations, you did not check the influence of 1RM on jumping performance ? I have serious concerns regarding the physiological explanation regarding the increased jumping height and buffer capacity, you hypothesized in the introduction to explain success in the FRAN.
Response:
We thank the reviewer for the comment. Yes, the correlations found between squat with CMJheight was r=0.629 and with CMJw/kg was r=0.659. Lower correlations almost to non-significant were found with the other strength exercises. In addition, no significant correlation was found between CJ variables and strength measurements. Consequently, we feel that the readers might confused by presenting these correlations into the text. Perhaps, due to the physiological demands of FRAN performance might base on other factors like the aerobic capacity of each athlete.
No Discussion was performed on the influence of maximal strength on strength-endurance capcity. If my maximal strength is very high, the used percentage strength value (e.g. 160kg you mentioned in the faster group) is smaller compared to the percentage of weaker athletes. Thus, it can also be hypothesized that they will show reduced fatigue and reduced accumulated lactate. Strength endurance performance is not only determined by the capacity to eliminate lactate, or resistvagainst high acidosis. If the needed percentage of the muscle does not trigger the anaerobic system, as the performance can be covered by the aerobic system no or lower lactate accumulation will arise. This can be explained by higher maximal strength. The Discussion (similar to the introduction) seems not physiologically guided, but accumulates several studies from the literature with questionable relevance for the study and needs a revision.
Response:
We thank the reviewer for the comment. We agree with the reviewer comment that stronger athletes may have an advantage in FRAN. We also would like to thank the reviewer for point us a possible explanation of our results. We added this explanation inside the discussion.
Additionally:
Lines 313 – 323: A descriptive approach and correlation between body fat and performance is no causal relationship. Just because athletes with lower body fat showed better performance, it is unvalid to state that the body fat will be the factor that caused this higher performance. Otherwise, every marathon running athlete with 2% body fat would outperform strength athletes.
Response:
We thank the reviewer for the comment. We agree with the reviewer that body fat may not be the only limiting factor in FRAN and we thank the reviewer for helping us in the specific point of the discussion. We changed this part according to the reviewer comment.
Lines 324 – 336: Same is true for CMJ height. Ignoring that CMJ and 1RM max strength might be highly correlated as well, as maximum strength commonly explains a large percentage of variance in the CMJ jump, using correlation coefficients for such statements in this section are very questionable.
Response:
We thank the reviewer for the comment. As we previously presented we found only moderate correlations between strength and CMJ. In addition, our findings are in line with a previous study (https://doi.org/10.3390/ijerph17103699) regarding the connection between CMJ and several crossfit workouts. We feel that the particular paragraph is not too provocative or challenging regarding the presentation of this correlation.
Lines 344 – 349: you state mixed sex distribution in previous studies as a limiting factor. Why? Does sex make the difference, if any exists? Hormonal status (there are women with high testosterone levels and men with low levels, other hormones (HGH; Insulin etc. with anabolic effects are present in both sexes). Did any study that performed sex difference analyses controlled hormonal differences? Just sex seems a week explanation, especially since there are numerous articles that show no significant difference in strength training adaptations between males and females.
Response:
We thank the reviewer for the question. The main problem when measuring 1-RM strength is the proportion of lean mass. Males have greater lean mass compare to females (https://doi.org/10.1152/jappl.2000.89.1.81; https://doi.org/10.3390/jfmk6010017) and this directly affect strength performance. In addition, looking the available scatterplots from the studies someone may notice the formation of clusters on the upper and lower parts of the trend-line when including both males and females. This directly affects the interpretation of the results. Consequently, we just mention this as a limitation from these studies.
Lines 351 – 356: The practical recommendations you state below might be correct (personally I would always agree with increasing 1Rm in the squat and focusing on Olympic lifts would be an essential part of every performance orientated or health related training routine), however, these recommendations might not be very well related to you study and it seems unvalid to state these based on your results. Please revise.
Response:
We thank the reviewer for the comment. We revised the practical recommendations.
I think there is much more potential in your data that should be outlines. Please increase the red line in the Introduction and Discussion by avoiding stating random crossfit studies, but focus the Introduction on the studies objective you investigated in the following. I would recommend to further include correlations between maximal strength and the other parameters (each parameter with each other) to get results for the Discussion that provide additional value, as you might not have to focus on correlations between CMJ and FRAN; but could also discuss the relationship of maximal strength in CMJ and FRAN together to formulate the outlook for further studies that should investigate the cross fit profile and relationships between different motoric functions, as well as checking the relevance of e.g. a deep squat training (lower extremity maximal strength), anaerobic training to resist against high lactate acids or buffer capacity, or isometric strength (IMTP) on specific cross fit performance.
Response:
We thank the reviewer for the comment and suggestions. We would like to take the opportunity here and kindly report to the reviewer that the references we included inside the manuscript are not random crossfit studies. On the contrary, these studies helped us to build our theory and compare our results. We also would like to avoid adding any other correlation results since this might confuse the readers and blur the overall message of the manuscript.
While results might be interesting in general, some more work has to be performed before I can recommend the publication of this manuscript. I hope some comments are helpful for your revisions. Looking forward to the revised version. Thank you very much for your work.
We thank the reviewer for all the comments kindly provided to us which help us to improve our manuscript. We hope that our revisions will satisfy the reviewer.
Reviewer 4 Report
Comments and Suggestions for Authors
Overall, I am pleased to review this paper and can recommend the publication after minor revisions. In general, the manuscript is fairly well-written and presents an interesting view of physiological parameter and the impact of the fitness of athletes on the performance of the FRAN WOD. Nevertheless, I think there are some points that, if improved, can help to improve the quality of the article. Below you will find my comments in detail.
Line 15. The authors wrote „30sec“. In the following also the term “seconds” is used. The authors should use one term thought out the manuscript in a uniform manner.
Line 32. The authors wrote “which are organized in training sessions called workout of the day”. In reference to the official CF training guide and current literature training sessions consist of different training part including the WOD [1-3]. In this sense, the statement should be corrected. Perhaps a definition of WOD would be helpful.
Line 142-144. The information provided on the content of FRAN may overlap with parts of the introduction. Authors should reduce redundancies.
Line 172 and 176. The authors used the term “min” in Line 172 and “minutes” in Line 176. See previous comment.
Line 257. The authors calculated the correlation including all athletes. Are there results on the correlation between the groups? How does the correlation change for the respective groups alone?
Line 320. Do the authors have a suggestion as to why this is the case? It could be discussed based on the existing data that the time of bodyweight workouts depends on the athletes' bodyweight. For example, it is known that female athletes in CF can achieve the same time in workouts with reduced external weight (e.g. dumbbells); however, this effect could not be observed in WOD with bodyweight exercises (e.g. pull-ups).
In addition, are there any results on which part of the WOD the time differences are higher? Which part (thrusters or pull-ups) impacts the overall time more? It would be interesting to investigate this in detail. In order to provide coaches and athletes with pacing strategies [4], it would be beneficial to analyze the data collected in this manuscript.
1. Dominski, F.H., R.A. Tibana, and A. Andrade, " Functional Fitness Training", CrossFit, HIMT, or HIFT: What is the preferable terminology? Frontiers in Sports and Active Living, 2022: p. 207.
2. Glassman, G., The CrossFit Training Guide. The CrossFit Journal, 2010. 9(9): p. 1-115.
3. Meier, N., D. Sietmann, and A. Schmidt, Comparison of Cardiovascular Parameters and Internal Training Load of Different 1-h Training Sessions in Non-elite CrossFit® Athletes. Journal of Science in Sport and Exercise, 2022: p. 1-12.
4. Mangine, G.T., et al., Workout Pacing Predictors of Crossfit Open Performance: A Pilot Study. Journal of Human Kinetics, 2021. 78(1): p. 89-100.
Author Response
Reviewer 4
Overall, I am pleased to review this paper and can recommend the publication after minor revisions. In general, the manuscript is fairly well-written and presents an interesting view of physiological parameter and the impact of the fitness of athletes on the performance of the FRAN WOD. Nevertheless, I think there are some points that, if improved, can help to improve the quality of the article. Below you will find my comments in detail.
Response:
We would like to thank the reviewer for taking the time to review our manuscript. We appreciate the efforts in providing us with comments that will enhance the quality of the paper and the message we are trying to deliver to the readership. We hope our replies and amendments are of satisfaction.
Line 15. The authors wrote „30sec“. In the following also the term “seconds” is used. The authors should use one term thought out the manuscript in a uniform manner.
Response:
We thank the reviewer for the correction.
Line 32. The authors wrote “which are organized in training sessions called workout of the day”. In reference to the official CF training guide and current literature training sessions consist of different training part including the WOD [1-3]. In this sense, the statement should be corrected. Perhaps a definition of WOD would be helpful.
Response:
We thank the reviewer for the comment. WOD was deleted from the manuscript according to the recommendations of reviewer 1.
Line 142-144. The information provided on the content of FRAN may overlap with parts of the introduction. Authors should reduce redundancies.
Response:
We thank the reviewer for the comment. We agree with the reviewer regarding the repeated information, but we would like to keep this in order to help the new to crossfit readers to better understand the workout. We hope that our decision will not disappoint the reviewer.
Line 172 and 176. The authors used the term “min” in Line 172 and “minutes” in Line 176. See previous comment.
Response:
We thank the reviewer for the correction.
Line 257. The authors calculated the correlation including all athletes. Are there results on the correlation between the groups? How does the correlation change for the respective groups alone?
Response:
We thank the reviewer for the intriguing comment. Correlations were almost identical between groups and the entire sample size. For example, the correlation between FRAN and %decrement of jumping height was r=-0.454 for n=20, r=-0.600 for fast group and r=-0.619 for slow. However, we think that presenting correlations for each group separately, will confuse the readers rather than help the interpretation of the results.
Line 320. Do the authors have a suggestion as to why this is the case? It could be discussed based on the existing data that the time of bodyweight workouts depends on the athletes' bodyweight. For example, it is known that female athletes in CF can achieve the same time in workouts with reduced external weight (e.g. dumbbells); however, this effect could not be observed in WOD with bodyweight exercises (e.g. pull-ups).
Response:
We thank the reviewer for the comment. We have changed the specific part in the manuscript according to the suggestions of the 3rd reviewer. Although body fat may have a limiting role in time-trial workouts, the conditioning of the neuromuscular system, the technique or the experience level of the athlete may also affect performance. From our perspective, we think that bodyweight exercises are much more difficult for female athletes leading to a slower pace during workouts. This is a really interesting question to investigate.
In addition, are there any results on which part of the WOD the time differences are higher? Which part (thrusters or pull-ups) impacts the overall time more? It would be interesting to investigate this in detail. In order to provide coaches and athletes with pacing strategies [4], it would be beneficial to analyze the data collected in this manuscript.
Response:
We thank the reviewer for the interesting comment. Indeed this would be helpful for establishing a pace during a workout. Regrettably, we only collected the total time during FRAN. However, we would like to thank the reviewer for pointing this to our attention. Perhaps in our next project we will include this interesting analysis, together with the one at the previous comment.
- Dominski, F.H., R.A. Tibana, and A. Andrade, " Functional Fitness Training", CrossFit, HIMT, or HIFT: What is the preferable terminology? Frontiers in Sports and Active Living, 2022: p. 207.
- Glassman, G., The CrossFit Training Guide. The CrossFit Journal, 2010. 9(9): p. 1-115.
- Meier, N., D. Sietmann, and A. Schmidt, Comparison of Cardiovascular Parameters and Internal Training Load of Different 1-h Training Sessions in Non-elite CrossFit® Athletes. Journal of Science in Sport and Exercise, 2022: p. 1-12.
- Mangine, G.T., et al., Workout Pacing Predictors of Crossfit Open Performance: A Pilot Study. Journal of Human Kinetics, 2021. 78(1): p. 89-100.
Reviewer 5 Report
Comments and Suggestions for Authors
Overall, your conclusion is not supported by your study. this is a cross-sectional study, and therefore you cannot state that faster Fran times are because the athletes have better body composition and better 1 rep max. while it may be intuitive, you still can't make that conclusions. maybe they have better 1rep max and CJ height because of faster Fran times- that direction can be equally as likely.
Other comments:
line 29: remove "famous" . I am not sure what "famous" means as a sport or training method.
line 36-37: this statement is not supported by your literature you sight. WODs are inherently scalable - and all fitness programs and sports have injury rates. I don't know what you mean by "properly designed." I would just remove this statement.
line 49: the study you reference used HIIT - with cycling. this is really NOT the same a a FRAN workout or some of the CrossFit workouts. while there are some HIIT in CrossFit - CrossFit is not always HIIT. you should remove the highlighted text.
line 119-121: state that those below the median were classified as "fast" and those above the median were classified as "slow"
line 277-278: remove the sentence "faster athletes.." this is a conclusion that you can't base on your study. you can hypothesize, but you already do that later, so just remove this sentence.
line 314: remove "it is evident that" - since you only base it on two studies and I don't think you can conclude this. "Studies support that systematic.." is a better way to start off.
Conclusion section
your conclusions need to be re-written. there are lots of editorial statements, and not supported by your study.
you can conclude that fitness parameters such as body comp, CMJ, 1 rep max are correlated with faster FRAN times.
that's it that you can conclude. try something like faster FRAN times may be predicted by body composition, CMJ and 1 rep thrusters - pick something that is easy to test.

Comments on the Quality of English Languageminor use of specific words. I recommended above.
Author Response
Reviewer 5
Overall, your conclusion is not supported by your study. this is a cross-sectional study, and therefore you cannot state that faster Fran times are because the athletes have better body composition and better 1 rep max. while it may be intuitive, you still can't make that conclusions. maybe they have better 1rep max and CJ height because of faster Fran times- that direction can be equally as likely.
Response:
We thank the reviewer for the initial comment. We also thank the reviewer for taking the time to review our manuscript.
We added in the limitations a sentence regarding the weaknesses that may arise form such a study design according to the comments of the first reviewer. Indeed, this is a cross-sectional study with two basic research questions. The study was pre-planned in order to investigate the differences between faster and slower athletes (see previous reference doi:10.3390/ijerph17103699), consequently, the reviewer’s suggestion regarding the reverse course of the research question and whether stronger or more powerful athletes may have a lower time-trial in FRAN, should be investigated from a future study.
Other comments:
line 29: remove "famous" . I am not sure what "famous" means as a sport or training method.
Response:
We thank the reviewer for the comment. We replaced famous with well-known.
line 36-37: this statement is not supported by your literature you sight. WODs are inherently scalable - and all fitness programs and sports have injury rates. I don't know what you mean by "properly designed." I would just remove this statement.
Response:
We thank the reviewer for the suggestion. The sentence was deleted.
line 49: the study you reference used HIIT - with cycling. this is really NOT the same a a FRAN workout or some of the CrossFit workouts. while there are some HIIT in CrossFit - CrossFit is not always HIIT. you should remove the highlighted text.
Response:
We thank the reviewer for the comment. We deleted the last words connecting HIIT with crossfit. The current reference helps us establish the rest paragraph. We hope that the reviewer will agree with our decision.
line 119-121: state that those below the median were classified as "fast" and those above the median were classified as "slow"
Response:
We thank the reviewer for the suggestion. The change has been made.
line 277-278: remove the sentence "faster athletes.." this is a conclusion that you can't base on your study. you can hypothesize, but you already do that later, so just remove this sentence.
Response:
We thank the reviewer for the suggestion. The sentence has been removed.
line 314: remove "it is evident that" - since you only base it on two studies and I don't think you can conclude this. "Studies support that systematic.." is a better way to start off.
Response:
We thank the reviewer for the suggestion. Change has been made.
Conclusion section
your conclusions need to be re-written. there are lots of editorial statements, and not supported by your study.
you can conclude that fitness parameters such as body comp, CMJ, 1 rep max are correlated with faster FRAN times.
that's it that you can conclude. try something like faster FRAN times may be predicted by body composition, CMJ and 1 rep thrusters - pick something that is easy to test.
Response:
We thank the reviewer for the suggestion. We followed the reviewer’s suggestion and re-wrote the conclusion section.
Round 2
Reviewer 2 Report
Comments and Suggestions for Authors
The question to the answer is unclear.
Author Response
Reviewer 2
The question to the answer is unclear.
Response:
We thank the reviewer for the comment. Unfortunately, we cannot locate the points inside the manuscript that the reviewer wants us to improve or change. This manuscript has received in total five (5) review reports and we have made substantial changes in all the paragraphs. However, if the reviewer feels that we must change something dramatically, then we would be more than happy to receive a third round of comments and suggestions. At this point, we would like to attest that we respect the reviewer for the efforts to improve our manuscript and we really looking forward to receiving potential new revisions.
Reviewer 5 Report
Comments and Suggestions for Authors
Line 24-25: Abstract - your last line is an overreaching conclusion of your article. it may predict - but doesn't necessarily mean that improving body comp will give you faster fran times.
Line 302: remove "base on their" and add "with"
Line 305: remove "strong"
Line 309-310 - again this is an overreach. and you repeat this later. I suggest removing this line.
Line 409-410: CrossFit programming does all this - so I'd remove this sentence
Line 411-413: would remove that all. again, overreach of the study. you can state that body comp and higher 1 rep max, and CJ, may predict faster Fran Times..but that's about it.
the bigger question to me is what is different training regiment or what do faster fran athletes do that is different than slower fran athletes besides their time training in CrossFit?
Comments on the Quality of English Language
some minor grammatical changes.
Author Response
Reviewer 5
Line 24-25: Abstract - your last line is an overreaching conclusion of your article. it may predict - but doesn't necessarily mean that improving body comp will give you faster fran times.
Response:
We thank the reviewer for the comment. We changed the last line of abstract as suggested.
Line 302: remove "base on their" and add "with"
Response:
We thank the reviewer for the correction.
Line 305: remove "strong"
Response:
We thank the reviewer for the suggestion. We removed the word.
Line 309-310 - again this is an overreach. and you repeat this later. I suggest removing this line.
Response:
We thank the reviewer for the comment. We removed the sentence.
Line 409-410: CrossFit programming does all this - so I'd remove this sentence
Response:
We thank the reviewer for the comment. We removed the sentence.
Line 411-413: would remove that all. again, overreach of the study. you can state that body comp and higher 1 rep max, and CJ, may predict faster Fran Times..but that's about it.
Response:
We thank the reviewer for the comment. We changed the sentence according to the reviewers suggestion.
the bigger question to me is what is different training regiment or what do faster fran athletes do that is different than slower fran athletes besides their time training in CrossFit?
Response:
We thank the reviewer for the intriguing question. We believe that there are no significant differences in training time or competition experience between fast and slow FRAN athletes. We believe, however, that there might be a deeper difference which would probably be a combination of pacing strategy during FRAN, differences in several biological determinants (faster twitch muscle fibres with lower thresholds, perhaps greater number of capillary vessels, greater anaerobic threshold etc), differences in the technique of the exercises and maybe different psychological treatment of the high intensity workout of FRAN. These would be interesting research questions for investigation.